# LARGE PRE-TRAINED TIME SERIES MODELS FOR CROSS-DOMAIN TIME SERIES ANALYSIS TASKS

## ABSTRACT

Large pre-trained models have been instrumental in significant advancements in domains like language and vision making model training for individual downstream tasks more efficient as well as provide superior performance. However, tackling time-series analysis tasks usually involves designing and training a separate model from scratch leveraging training data and domain expertise specific to the task.

We tackle a significant challenge for pre-training a general time-series model from multiple heterogeneous time-series dataset: providing semantically useful inputs to models for modeling time series of different dynamics from different domains. We observe that partitioning time-series into segments as inputs to sequential models produces semantically better inputs and propose a novel model LPTM that automatically identifies optimal dataset-specific segmentation strategy leveraging self-supervised learning loss during pre-training.

LPTM provides performance similar to or better than domain-specific state-of-art model and is significantly more data and compute efficient taking up to 40% less data as well as 50% less training time to achieve state-of-art performance in a wide range of time-series analysis tasks from multiple disparate domains.

## 1 INTRODUCTION

Time-series analysis tasks involve important well-studied problems involving time-series datasets such as forecasting (Hyndman & Athanasopoulos, 2018) and classification (Chowdhury et al., 2022) with applications in wide-ranging domains such as retail, meteorology, economics, and health. Recent works (Chen et al., 2021; Wang et al., 2022; Zeng et al., 2023) have shown the efficacy of purely data-driven deep learning models in learning complex domain-specific properties of the time series over traditional statistic and mechanistic models across many domains.

However, coming up with a model for a specific application or time-series analysis task is usually non-trivial. Most state-of-art neural models are known to be data-hungry and require substantial training data from the same domain on which we deploy to train. This can be prohibitive in many real-world applications. While we have access to a large amount of time-series datasets from other tasks and domains, that contain useful background patterns and information, time-series models typically cannot leverage them to improve their performance.

In contrast, for many language and vision tasks, we use pre-trained models trained on a larger *pre-training dataset* (Qiu et al., 2020; Du et al., 2022; Gunasekar et al., 2023). These pre-trained models are then fine-tuned to the downstream task. There are two important benefits of pre-trained models. First, the pre-trained weights are good initialization for faster and more effective training. They require less training resources and data and produce superior performance. Moreover, pre-trained models learn useful underlying structures and patterns from larger pre-trained datasets such as common syntactic and semantic knowledge in the case of language and the ability to recognize useful patterns in the case of vision. Initiating training from these pre-trained models usually results in faster training and better performance. compared to training the model from scratch on task-specific training data.

Therefore, we tackle the goal of building a unified pre-trained models for time-series that are pre-trained on datasets from multiple domains and can be applied to a wide range of downstream time-series analysis tasks across all domains. However, there are important challenges intrinsic to

time-series that makes pre-training non-trivial. Most neural sequential models input time-series values for each time-step separately. However, unlike text data, each individual time stamp may not provide enough semantic meaning about local temporal patterns of the time series. To tackle this, Nie et al. (2022) proposed to segment the time series and input each segment as individual tokens to their transformer-based model and showed superior performance to more complex transformer-based architectures. However, in the case of pre-training with multiple domains, each dataset in pre-train datasets are derived from different domains with different set of underlying generative dynamics, sampling rate, noise, etc. Using uniform segment sizes similar to Nie et al. (2022) for all datasets would be suboptimal. For example, among two datasets, a dataset with a higher sampling rate may require longer segments than those with lower sampling rates to capture similar patterns in the model. Further, the optimal segment size used for the same time-series may vary with time. For, time intervals that are smoother with less complex dynamics, using longer segment sizes may suffice whereas intervals where time-series have more complex and multiple temporal patterns may require finer-grained segmentation.

We tackle these challenges and propose **Large Pre-trained Time-series Models** (LPTM), a novel method for generating pre-trained models for time-series data across multiple domains. LPTM uses a simple transformer-based architecture and leverages a self-supervised pre-training to simultaneously train on multiple datasets from different domains. We utilize simple self-supervised tasks based on masking tokens input to the transformer and learning to reconstruct the masked tokens. However, we input segments of time-series as tokens to the transformer. To overcome the challenges associated with segmentation on diverse datasets discussed above, we propose a novel adaptive segmentation module that segments the time-series of each domain based on how well it performs on self-supervised pre-training. The segmentation module uses a novel scoring mechanism for the segmentation strategy used by the model on input time-series for a domain based on the SSL (self-supervised learning) loss and optimize the segmentation strategy to lower the SSL loss. We show that LPTM can be fine-tuned to a variety of forecasting and classification tasks in varied domains such as epidemiology, energy, traffic, economics, retail, and behavioral datasets. We also show that LPTM can provide performance on par with state-of-art models with lesser training data during fine-tuning as well as with fewer training steps showcasing the efficiency of our pre-trained framework. Our main contributions can be summarized as follows:

1. **Multi-domain Pre-trained time-series model** We propose a framework for generating large pre-trained models for time-series that are trained on multiple datasets across varied domains. LPTM is an important step towards general pre-trained models for time-series similar to LLMs for text and vision.

2. **Adaptive segmentation for cross-domain pre-training** To optimally extract semantically useful information from time-series of different domains with varied dynamics and sampling rates for pre-training, we propose a novel adaptive segmentation module that learns segmentation strategy for each domain based on losses from self-supervised learning tasks.

3. **State-of-art and efficient performance in diverse downstream time-series tasks** We evaluate LPTM on downstream forecasting and classification tasks from multiple domains and observe that LPTM consistently provides performance similar to or better than previous state-of-art models usually using lesser training steps and compute time. We also observe that LPTM typically requires less than 80% of training data used by state-of-art baselines to provide similar performance.

## 2 PROBLEM SETUP

**Time-series analysis tasks** Our pre-trained model can be used for many time-series tasks including forecasting and classification from multiple benchmarks and domains. For a given downstream task let $\mathcal{D}^T$ be the time-series dataset consisting of time series $\mathbf{y}^{1...T}$. A time-series analysis task's goal is to predict important properties of the time-series. For example, the forecasting task involves predicting the future values $\mathbf{y}^{T+1...T+K}$ whereas classification involves predicting the class label of the input time-series based on labeled training data.

**Self-supervised pre-training on multi-domain datasets** The goal of our work is to learn useful knowledge and patterns from time-series datasets from time-series from different domains. This

is in contrast to previous works which typically train the models only on time-series from current downstream tasks.

Formally, we have access to time-series datasets from $K$ domains where the datasets of domain $k$ is denoted as $\mathcal{D}'_k = \{\mathcal{D}'_{k,i}\}_{i=1}^{N(k)}$ where $N(k)$ is the number of datasets in domain $k$. Examples of these domains include epidemiology, energy forecasting, macroeconomics, traffic prediction, etc. The entire set of heterogenous multi-domain *pre-train* dataset is denoted as $\mathcal{D}_{\text{pre}} = \{\mathcal{D}'_1, \mathcal{D}'_2, \ldots, \mathcal{D}'_K\}$. In order to effectively pre-train LPTM on $\mathcal{D}_{\text{pre}}$ we formulate the problem as a set of self-supervised learning tasks $\mathcal{T}_{\text{pre}} = \{\mathcal{T}_i\}_{i=1}^{R}$ on the set of pre-training datasets $\mathcal{D}_{\text{pre}}$. During pre-training, we sample $(\mathcal{D}'_{k,i}, k)$, a dataset and its domain label from $\mathcal{D}_{\text{pre}}$ and train the model on each of the self-supervised learning tasks in $\mathcal{T}_{\text{pre}}$. The tasks in $\mathcal{T}_{\text{pre}}$ are self-supervised and do not require additional labels or other ground truth. These tasks transform the input data and train the model to recover the original input or important properties or parts of the input.

Therefore, our problem can be formally stated as: *Given a heterogeneous set of multi-domain datasets $\mathcal{D}_{pre}$ and their domain labels, we train a model leveraging SSL tasks $\mathcal{T}_{pre}$ that learns important patterns and knowledge that can be leveraged on fine-tuning the model to any time-series analysis task on any novel dataset from any of the domains $d \in \{1, 2, \ldots, K\}$.*

## 3 METHODOLOGY

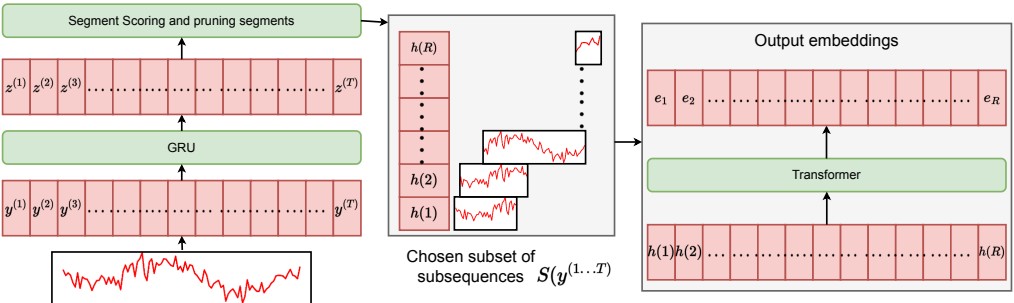

Figure 1: Overview of LPTM. The input time-series $y^{(1\ldots T)}$ is first segmented based on a scoring function optimized using SSL loss. The segments are fed as individual tokens to the transformer encoder to get output embeddings of time-series that are used for downstream tasks.

### 3.1 OVERVIEW

Similar to model piplelines used in NLP and vision, we first train a pre-trained model $M(\theta_{pre})$ on multiple pre-training datasets $\mathcal{D}_{\text{pre}}$. Most of the parameters $\theta_{pre}$ of the pre-trained model are trained over all the datasets and tasks. However, we use a separate segmentation module for each dataset domains to capture varied sizes of segments that differ across datasets. These segments are used as tokens for a transformer model that shares the parameters across all the tasks. For each of the pre-trained task as well as downstream tasks we append a final linear layer on the output embeddings of the transformer to generate the final prediction. Note that during fine-tuning on downstream tasks we update the parameters of all the modules of LPTM.

### 3.2 SEGMENTATION MODULE

Due to their ability to model long-range temporal relations as well as scale up to learn from large datasets, transformers (Vaswani et al., 2017) are increasingly used for time-series tasks. Recent works Zhou et al. (2021; 2022); Liu et al. (2021); Chen et al. (2021) have shown the efficacy of transformers for time-series forecasting in a wide range of domains.

Previous works input each time-step of a time-series as individual tokens. Unlike text, individual time-steps do not typically provide any semantic meaning about the temporal patterns of the time-series. Therefore, Nie et al. (2022) proposed to segment the input time-series into uniform length

segments and use each of the segments as tokens to the transformer model. However, different pre-trained datasets may have varied temporal scales, periodicity and other temporal dynamics that cannot be encompassed by a single uniform segmentation strategy. For example, epidemic time-series are usually observed at weekly scale and may have characteristic properties like seasonality, peaks and sudden outbreaks that should be captured by segmentation. Economic time-series, in contrast, are typically captured every quarter and are more monotone with sudden anomalies and changes in data distribution. Moreover, using a uniform segmentation may not be ideal for time series that have multi-scale trends with some time-stamps having denser temporal information requiring finer-graned segmentation than others. Therefore, our goal is to identify an independent segmentation strategy for each domain of time-series dataset.

For a given input time-series $\mathbf{y}^{(1...t)}$, we pass it through a GRU to get hidden embeddings $\{\mathbf{z}^{(i)}\}_{i=1}^{t}$ that models the temporal patterns of the input:

$$\{\mathbf{z}^{(i)}\}_{i=1}^{t} = \text{GRU}_1(\{y^{(i)}\}_{i=1}^{t}). \tag{1}$$

We then introduce a *segment score function s* that provides a scalar score for any subsequence of the input time-series:

$$s(i,j) = \mathbf{v}^T \tanh\left(\mathbf{W}_1 z_i + \mathbf{W}_1 z_j + \mathbf{b}\right). \tag{2}$$

The score $s(i,j)$ for a subsequence from time-stamp $i$ to $j$ denotes how good the given segment is for the dataset.

In next step, we sample subset $S(y^{(1...t)})$ of subsequences over the time-series that a) covers the entire input time-series, b) has a high score function value. While retrieving the optimal $S(y^{(1...t)})$ is an interesting combinatorial optimization problem, we generate $S(y^{(1...t)})$ using a simple process as follows: for each $i \in \{1, 2, \ldots, t-1\}$, we denote $h(i) = \arg\max_{j \in \{i+1..., t-1\}} s(i,j)$ as the best segment starting from time-step $i$. Then we generate the set of segments $\hat{S}(y^{(1...t)}) = \{(i, h(i))\}_{i=1}^{t-1}$. In order to reduce the number of segments, we iteratively remove the lowest-scoring segments until we cannot remove any more segments without having time-steps not being covered by any segments in the set. The final set of segments after pruning is denoted as $S(y^{(1...t)})$.

To generate the token embeddings for each segment $(i,j)$, we pass the embeddings $\{z^{(i)}, z^{(i+1)}, \ldots, z^{(j)}\}$ through a self-attention layer used in transformers and aggregate the output embeddings. Additionally, we concatenate the following features to the token embedding of each segment token:

- Positional encoding of the starting time-step of the segment $pos(i)$ defined as:

$$\text{pos}(i) = \begin{cases} \sin(i/10^{5i/D}) & \text{if } i \text{ is even} \\ \cos(i/10^{5(i-1)/D}) & \text{if } i \text{ is odd.} \end{cases} \tag{3}$$

  where $D$ is the dimensions of output embedding of self-attention over $\{e_i, e_{i+1}, \ldots, e_j\}$.

- Positional encoding of the length of the segment $pos(j-i)$

- The time-series values of segment are passed though a single layer of transformer encoder and aggregated to a fixed length embedding of dimension $D$.

These features allow the transformer additional information from the segment directly derived from values of time-series. The final output of the segmentation module is a sequence $\{e_i\}_{i=1}^{R}$ where $R$ is the size of $S(y^{(1...t)})$ and sequence is arranged based on the ascending order of the first time-stamp of each segment.

## 3.3 SELF-SUPERVISED LEARNING TASKS

Pre-training on a wide range of heterogeneous datasets from multiple domains helps LPTM learn from useful patterns and latent knowledge across these domains that can be generalized to range downstream tasks on multiple domains. We propose two general self-supervised learning tasks motivated by pre-trained language models to enable LPTM to learn from all pre-trained datasets. We leverage a transformer model and use the segment token embeddings of the segmentation module. The two pre-training SSL tasks are **Random Masking** (RANDMASK) and **Last token masking** (LASTMASK). RANDMASK allows the model to extrapolate and interpolate masked segments of

the input time-series. RANDMASK has also been explored for representation learning in previous works (Zerveas et al., 2021; Nie et al., 2022) but they are applied on the same dataset as that used for training unlike our data and task-agnostic pre-training setup. Formally, we mask each input segment token with a probability of $\gamma$ and decode the values of time-series of the masked segments from the output embeddings of the transformer. We use a simple GRU with a single hidden layer on the transfer's output embedding to decode the values of the segment and use mean-squared error as the loss. LASTMASK is similar to RANDMASK except we mask last $\gamma$ fraction of the segments. This allows the model to forecast the future values of the time-series, a very important task in many time-series domains.

## 3.4 TRAINING DETAILS

**Instance normalization**   The values of the time-series of each dataset can vary widely based on the application the the target value observed in the time-series. Therefore, as part of pre-processing we first normalize the time-series of each dataset of pre-train datasets independently. Moreover, the data distribution and the magnitude of the time-series can vary across time. We use reversible instance normalization (REVIN) layer Kim et al. (2021). REVIN performs instance normalization on the input time-series and reverses the normalization of the output values. The normalization step is part of the neural model and gradients are calculated over the normalization and reverse normalization layers.

**Training the score function**   We use the loss from the SSL tasks to also train the score function of the segmentation module. Since there is no direct gradient flow between the score function and the final predictions, due to the discrete nature of choosing the segments, we match the aggregated scores of all the chosen segments in $S(y^{(1...t)})$ to the negative logarithm of the total MSE loss of both SSL tasks:

$$\mathcal{L}_g = \left( \sum_{(i,j) \in S(y^{(1...t)})} g(i,j) + \log(\mathcal{L}_{SSL}) \right) \tag{4}$$

where $\mathcal{L}_{SSL}$ is the total loss of both SSL tasks. We also backpropagate over $\mathcal{L}_g$ once every 10 batches. This is to stabilize training since changing the segmentation strategy for every batch leads to unstable and inefficient training.

**Linear-probing and fine-tuning**   Kumar et al. (2022) showed that fine-tuning all the parameters of the pre-trained model for a specific downstream task can perform worse than just fine-tuning only the last layer (linear probing), especially for downstream tasks that are out-of-distribution to pre-trained data. To alleviate this, based on the recommendation from Kumar et al. (2022), we perform a two-stage fine-tuning process: we first perform linear probing followed by fine-tuning all the parameters.

## 4   EXPERIMENT SETUP

### 4.1   DATASETS

We derive pre-train time-series datasets from multiple domains:

1. **Epidemics:** We use a large number of epidemic time-series aggregated by Project Tycho (van Panhuis et al., 2018). from 1888 to 2021 for different diseases collected at state and city levels in the US. We remove time series with missing data and use time series for 11 diseases of very diverse epidemic dynamics such as seasonality, biology, geography, etc.: Hepatitis A, measles, mumps, pertussis, polio, rubella, smallpox, diphtheria, influenza, typhoid and Cryptosporidiosis (Crypto.).

2. **Electricity:** We use ETT electricity datasets (ETT1 and ETT2) collected from (Zhou et al., 2021) at 1 hour intervals over 2 years. We use the default 12/4/4 train/val/test split and use the train split for pre-training as well.

3. **Traffic Datasets:** We use 2 datasets related to traffic speed prediction. PEMS-Bays and METR-LA (Li et al., 2017) are datasets of traffic speed at various spots collected by the

Los Angeles Metropolitan Transportation Authority and California Transportation Agencies over 4-5 months.

4. **Demand Datasets:** We use bike and taxi demand datasets from New York City collected from April to June 2016 sampled every 30 minutes. We all but the last 5 days of data for training and pre-training.

5. **Stock forecasting**: We also collect the time-series of daily stock prices of Nasdaq and S&P 500 index using `yfinance` package (yfi) from July 2014 to June 2019. We train and pre-train using the first 800 trading days and use the last 400 for testing.

6. **M3 competition time-series**: We also used the 3003 time-series of M3 forecasting competition (Makridakis & Hibon, 2000) which contains time-series from multiple domains including demographics, finance, and macroeconomics.

7. **Motion and behavioral sensor datasets**: We use the set of sensor datasets extracted from UEA archive (Bagnall et al., 2018) and UCI Machine learning repository (Asuncion & Newman, 2007) similar to (Chowdhury et al., 2022).

## 4.2 Downstream tasks

We test the pre-trained LPTM trained on datasets discussed in §4.1 on multiple forecasting and time-series classification tasks. We perform forecasting on the influenza incidence time series in US and Japan. Specifically, we use the aggregated and normalized counts of outpatients exhibiting influenza-like symptoms released weekly by CDC[1]. For influenza in Japan, we use influenza-affected patient counts collected by NIID[2]. We forecast up to 4 weeks ahead over the period of 2004 to 2019 flu seasons using a similar setup as Flusight competitions Reich et al. (2019).

We also perform electricity forecasting on the ETT1 and ETT2 datasets using the train/test split mentioned previously. The last 10% of PEM-Bays dataset is used for traffic forecasting up to 1 hour ahead and the last 5 days of New York demand datasets for demand forecasting up to 120 minutes in the future. We also perform forecasting on the Nasdaq dataset for up to 5 days ahead and M3 time-series for 1 month ahead. We use 6 of the sensor datasets from Asuncion & Newman (2007) for time-series classification tasks. We use an 80-20 train-test split similar to Chowdhury et al. (2022).

## 4.3 Baselines

We compared LPTM's performance in a wide range of time-series tasks against seven state of art general forecasting baselines as well as domain-specific baselines. We compared with (1) Informer Zhou et al. (2021) and (2) Autoformer Chen et al. (2021), two state-of-the-art transformer-based forecasting models. We also compare against the recent model (3) MICN (Wang et al., 2022) which uses multiple convolutional layers to capture multi-scale patterns and outperform transformer-based models. We also compared against best models for individual tasks for each domain. For influenza forecasting, we compared against previous state-of-art models (4) EpiFNP Kamarthi et al. (2021) and (5) ColaGNN Deng et al. (2020) respectively. We also compare against (6) STEP Shao et al. (2022) that leverages Graph Neural Networks for forecasting and provides the best performance for demand forecasting, traffic prediction, and stock prediction benchmarks among the baselines by automatically modeling sparse relations between multiple features of the time-series. For classification tasks on behavioral datasets, we compare against the state-of-art performance of (7) TARNet Chowdhury et al. (2022).

In order to test the efficacy of our multi-domain pre-training method, we also compare it against two other state-of-art self-supervised methods for time-series. These prior SSL methods (Yue et al., 2022; Tonekaboni et al., 2021; Eldele et al., 2021; Nie et al., 2022) have shown to improve downstream performance by enabling better representation learning. However, the SSL pre-training is only done on the same dataset used for training for the downstream task and does not cater to pre-training on multiple heterogenous datasets from varied domains, unlike LPTM. Therefore, we also compare LPTM against previous works on self-supervised representation learning on time-series: TS2Vec (Yue et al., 2022) and TS-TCC (Eldele et al., 2021).

---

[1]https://gis.cdc.gov/grasp/fluview/fluportaldashboard.html
[2]https://www.niid.go.jp/niid/en/idwr-e.html

# 5 RESULTS

The code for implementation of LPTM and datasets are provided at anonymized link[3] and hyperparameters are discussed in the Appendix.

## 5.1 FORECASTING AND CLASSIFICATION TASKS

We summarize the forecasting performance using RMSE scores in Table 1. LPTM is either the first or a close second best-performing model in all the benchmarks in spite of comparing our domain-agnostic method against baselines designed specifically for the given domains. LPTM beats the previous state-of-art domain-specific baselines in five of the benchmarks and comes second in four more. Moreover, LPTM improves upon the state-of-art on electricity forecasting, traffic forecasting, and M3 datasets. Further, we observe that LPTM is better than other transformer-based state-of-art general time-series forecasting models as well as SSL methods which underperform all other baselines in most cases. This, therefore, shows the importance of our modeling choices to be capable of learning from diverse time-series datasets to provide performance that is similar to or better than previous state-of-art in most downstream tasks.

Table 1: Average forecast performance (measured as RMSE over 10 runs) of LPTM and baselines over different domains. The best model is in **bold** and the second best is underlined.

| Model | Flu-US | Flu-japan | ETT1 | ETT2 | PEM-Bays | NY-Bike | NY-Taxi | Nasdaq | M3 |
|---|---|---|---|---|---|---|---|---|---|
| Informer | 1.62 | 1139 | 0.57 | 0.71 | 3.1 | 2.89 | 12.33 | 0.83 | 1.055 |
| Autoformer | 1.41 | 1227 | 0.72 | 0.82 | 2.7 | 2.73 | 12.71 | 0.19 | 0.887 |
| MICN | 0.95 | 1145 | **0.49** | 0.57 | 3.6 | 2.61 | 11.56 | 0.13 | 0.931 |
| STEP | 1.17 | 983 | 0.54 | 0.93 | 2.7 | 2.52 | **10.37** | **0.11** | 1.331 |
| EpiFNP | **0.52** | 872 | 0.81 | 1.25 | 4.1 | 2.98 | 12.11 | 0.28 | 1.281 |
| ColaGNN | 1.65 | **694** | 0.72 | 1.19 | 3.9 | 3.19 | 14.97 | 0.25 | 1.185 |
| TS2Vec | 1.85 | 905.9 | 0.99 | 1.74 | 3.5 | 3.11 | 13.48 | 0.94 | 1.344 |
| TS-TCC | 1.94 | 1134.6 | 0.75 | 1.29 | 3.3 | 2.97 | 15.55 | 0.76 | 1.274 |
| LPTM | 0.79 | 704 | **0.49** | **0.46** | **2.5** | **2.37** | 11.84 | 0.17 | **0.872** |
| LPTM-NoSegment | 0.93 | 766 | 0.57 | 0.55 | 3.2 | 3.17 | 14.96 | 0.27 | 1.146 |
| LPTM-NoPreTrain | 0.96 | 827 | 0.46 | 0.57 | 3.7 | 2.66 | 12.43 | 0.25 | 1.271 |
| LPTM-NoLinProb | 0.92 | 885 | 0.43 | 0.53 | 3.1 | 2.49 | 12.17 | 0.19 | 1.032 |

Table 2: Average classification performance (measured as F1 score over 10 runs) of LPTM and baselines over different domains. The best model is in **bold** and the second best is underlined. The best model is statistically significant over the baselines ($p \leq 0.05$) when it beats the previous state-of-art.

| | BasicMotions | FaceDetection | FingerMovements | PEMS-SF | RacketSports | EigenWorms |
|---|---|---|---|---|---|---|
| Informer | 0.95 | 0.51 | 0.58 | 0.67 | 0.83 | 0.49 |
| Autoformer | 0.93 | 0.49 | 0.54 | 0.71 | 0.86 | 0.62 |
| TARNet(SOTA) | **1.00** | 0.63 | 0.62 | **0.94** | **0.98** | 0.89 |
| TS2Vec | 0.99 | 0.51 | 0.46 | 0.75 | 0.77 | 0.84 |
| TS-TCC | **1.00** | 0.54 | 0.47 | 0.73 | 0.85 | 0.77 |
| LPTM | **1.00** | **0.79** | **0.78** | 0.93 | 0.93 | **0.94** |
| LPTM-NoSegment | 0.98 | 0.68 | 0.57 | 0.66 | 0.66 | 0.59 |
| LPTM-NoPreTrain | 0.96 | 0.74 | 0.62 | 0.79 | 0.79 | 0.63 |
| LPTM-NoLinProb | **1.00** | 0.79 | 0.69 | 0.89 | 0.93 | 0.92 |

We evaluate LPTM and baselines on the classification of sensor and behavioral datasets from (Asuncion & Newman, 2007). We report the F1 scores in Table 2. We observe that LPTM outperforms the previous state-of-art model, TARNet (Chowdhury et al., 2022) in 3 datasets and is a close second best model in others.

---

[3]https://anonymous.4open.science/r/SegmentTS-6145/

## 5.2 DATA EFFICIENCY

A significant advantage of leveraging pre-trained models in the case of vision and language models is that we do not require a large amount of training data for fine-tuning to a specific task. In fact, in many cases, we require very few examples (Brown et al., 2020) to fine-tune the model.

We evaluate the efficacy of LPTM to train with a smaller fraction of task-specific training data. For each time-series analysis task, we fine-tune the model using only $k\%$ of training data for different values of $k$. The $k\%$ chosen is generated by using on the first $k\%$ of the timestamps' values. We do not choose a random sample to prevent data mixing from the rejected portion of training data. We also performed the similar experiment on the best baseline for each task and compare data efficiency of baseline with LPTM.

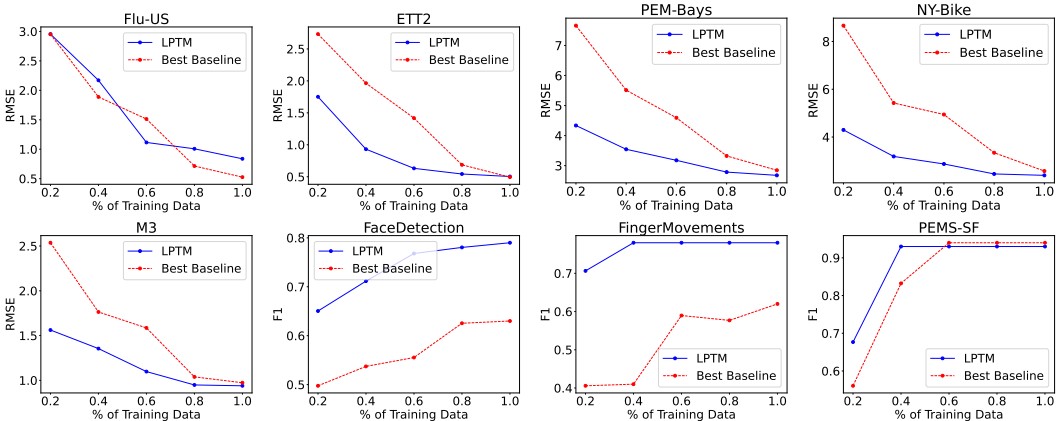

Figure 2: Performance of LPTM and best baseline with varying fractions of training data. In most cases LPTM significantly outperforms baselines with lower amount of data.

The comparison plots are shown in Figure 2. With lesser data, the performance of the baseline is much worse whereas LPTM typically requires much less data to provide similar performance to when we have access to the full dataset. This shows the importance of pre-training to quickly ramp up the performance of the model with much less data, a problem we encounter is many real-world settings such as when we need to deploy a forecasting model on novel applications such as a new pandemic with sparse data availability.

## 5.3 TRAINING EFFICIENCY

Another important advantage of pre-trained models is that they require much less training time and resources to fine-tune to a downstream task compared to time required for pre-training or even training from scratch. We compare the training (or fine-tuning) time of LPTM with baselines on benchmarks from different domains. We also measure the avergae time required by LPTM to reach the performance of best baseline in cases where we eventually outperform them.

The training times are summarized in Table 3. First, we observe that the time taken by LPTM to reach the performance of best best-performing baseline (LPTM-TB) is significantly smaller than the time taken by any other baselines. Further, even in cases where LPTM doesn't outperform the best baseline, it typically converges much faster. This shows that LPTM requires fewer training steps and therefore less compute time to fine-tune to any downstream task.

## 5.4 ABLATION STUDY

We finally study the impact of our various technical contributions to LPTM by performing an ablation study. Specifically, we formulate the following variants of LPTM to study the impact of our important modeling choices:

- LPTM-NoSegment: We remove the novel segmentation module and directly encode each time-step as a separate token.

Table 3: Average training time (minutes) till convergence for LPTM and baselines. LPTM-TB shows the time taken by LPTM to reach performance of top baseline (in benchmarks where LPTM outperforms it). Since some baselines are specific to forecasting or classification and we do not beat the state-of-art in few benchmarks we designate these cells in the table as NA.

| Model | Flu-US | ETT2 | PEM-Bays | NY-Bike | Nasdaq | M3 | BasicMotions | EigenWorms |
|---|---|---|---|---|---|---|---|---|
| Informer | 27.3 | 25.5 | 45.1 | 49.7 | 27.1 | 49.6 | 17.5 | 14.3 |
| Autoformer | 19.5 | 29.3 | 49.5 | 55.2 | 18.5 | 45.1 | 11.9 | 19.7 |
| MICN | 17.6 | 15.7 | 39.7 | 41.1 | 19.2 | 33.9 | NA | NA |
| STEP | 25.4 | 34.1 | 52.7 | 74.3 | 29.7 | 52.8 | NA | NA |
| EpiFNP | 22.5 | 39.5 | 41.1 | 39.1 | 21.6 | 97.6 | NA | NA |
| ColaGNN | 34.7 | 33.6 | 53.1 | 47.6 | 32.1 | 72.2 | NA | NA |
| TARNet | NA | NA | NA | NA | NA | NA | 13.7 | 9.4 |
| TS2Vec | 29.3 | 28.2 | 41.9 | 41.9 | 29.8 | 67.4 | 9.3 | 13.2 |
| TS-TCC | 21.7 | 23.7 | 46.3 | 44.3 | 25.3 | 55.8 | 12.7 | 11.1 |
| LPTM | **12.2** | 19.3 | 41.9 | **37.5** | **17.3** | **31.2** | **6.1** | 12.7 |
| LPTM-TB | NA | **12.5** | **29.6** | **32.9** | NA | **23.7** | **6.1** | **8.1** |

- LPTM-NoPreTrain: We do not perform any pre-training and instead directly learn from scratch for each downstream task.

- LPTM-NoLinProb: Instead of the two-step fine-tuning procedure discussed in §3.4, where we first fine-tune only the last layer (linear-probing) followed by fine-tuning all parameters of the model, we skip the linear-probing.

The performance of the ablation variants for forecasting and classification tasks are also shown in Tables 1 and 2 respectively. We observe that the ablation variants' performances are significantly worse than the variants, underperforming some of the baselines. The worst performing variant is usually LPTM-NoSegment, showing the importance of deriving good time-series segments to improve representation learning of time-series for each dataset.

## 6 CONCLUSION

We make a significant contribution towards general pre-trained models for time-series analysis tasks replicating the success of large pre-trained models in language and vision domains. We introduce LPTM, a general pre-trained model that provides state-of-art performance on a wide range of forecasting classification tasks from varied domains and applications. LPTM provides similar performance to state-of-art domain-specific models in applications such as epidemiology, energy, traffic, and economics and significantly beats state-of-art in widely used traffic prediction and M3 datasets. We also observe that LPTM required significantly lesser training data during fine-tuning to reach optimal performance compared to other baselines in most benchmarks. LPTM is also more efficient by requiring much less training steps (20- 50% lesser) to attain similar performance as domain-specific models.

Our work mainly focuses on the important challenge of providing semantically meaningful inputs to the model that caters to learning time-series segmentation strategies specific to each domain. This is crucial when pre-training on diverse datasets, a key challenge for time-series data. The underlying model architecture is a straightforward transformer encoder that uses well-known masking techniques for self-supervised pre-training. Therefore, our method can be extended to leverage novel time-series model architectures and SSL methods. Extending our methods to provide calibrated forecasts that provide reliable uncertainty measures is also another important direction of research.

Since our model can be applied to any generic time-series analysis tasks including those in critical domains such as public health, medicine, economics, etc., important steps need to be taken to address potential misuse of the our methods such as testing for fairness, data quality issues, ethical implications of predictions, etc.

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
