## A   RELATED WORKS

**Neural models for time-series analysis**   DeepAR Salinas et al. (2020) is a popular forecasting model that trains an auto-regressive recurrent network to predict the parameters of the forecast distributions. Deep Markov models Krishnan et al. (2017); Rangapuram et al. (2018); Li et al. (2021); Gu et al. (2021) model the transition and emission components with neural networks. Recent works have also shown the efficacy of transformer-based models on general time-series forecasting Oreshkin et al. (2019); Zhou et al. (2021); Chen et al. (2021); Zhou et al. (2022); Liu et al. (2021). However, these methods do not perform pre-training and are trained independently for each application domain. therefore, they do not leverage cross-domain datasets to generate generalized models that can be used for a wide range of benchmarks and tasks.

**Self-supervised learning for time-series**   Recent works have shown the efficacy of self-supervised representation learning for time-series for various classification and forecasting tasks in a wide range of applications such as modeling behavioral datasets Merrill & Althoff (2022); Chowdhury et al. (2022), power generation Zhang et al. (2019), health care Zhang et al. (2022). Franceschi et al. (2019) used triplet loss to discriminate segments of the same time-series from others. TS-TCC used contrastive loss with different augmentations of time-series Eldele et al. (2021). TNC Tonekaboni et al. (2021) uses the idea of leveraging neighborhood similarity for unsupervised learning of the local distribution of temporal dynamics. TS2Vec leveraged hierarchical contrastive loss across multiple scales of the time-series Yue et al. (2022). However, all these methods apply SSL on the same dataset that is used for training and may not adapt well to using time-series multiple sources such as time-series from multiple diseases. Our work, in contrast, tackles the problem of learning general models from a wide range of heterogeneous datasets that can be fine-tuned for a wide variety of tasks on multiple datasets that may not be used during pre-training.

## B   TRAINING DETAILS

For GRU we use a single hidden layer of 50 hidden units. Dimension of $\mathbf{v}$ is also 50. The transformer architecture consists of 6 layers with 8 attention heads each. For forecasting tasks, we train a separate decoder module with 4 more layers during fine-tuning whereas for classification we aggregate the embeddings $\{e_i\}_{i=1}^{R}$ of the last transformer layer and feed them into a single linear layer that provides logits for all classes. The SSL pre-training was done till convergence via early stopping with a patience of 1000 epochs. We observed that LPTM takes 5000-8000 epochs to finish pre-training which takes around 3-4 hours. (Note that pre-training is a one-time step and downstream fine-tuning takes much less time and epochs). For both pre-training and fine-tuning, we used the Adam optimizer with a learning rate of 0.001. The hyperparameters are tuned sparingly for both LPTM and baselines from their default settings. For RANDMASK, we found the optimal $\gamma = 0.4$, and for LASTMASK $\gamma = 0.2$ was optimal. The model was trained on a Nvidia Tesla V100 GPU with 32 GB memory.

# C  DATA EFFICIENCY

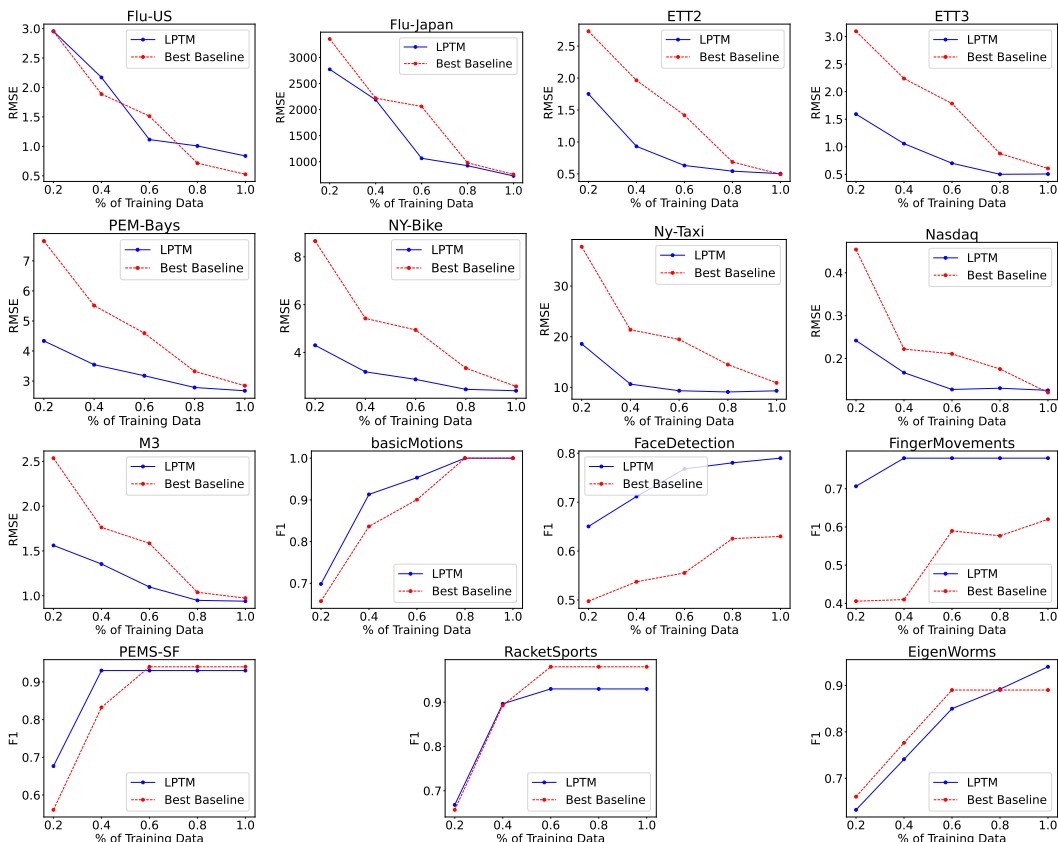

Figure 3: Performance of LPTM and best baseline with varying fractions of training data. In most cases LPTM significantly outperforms baselines with lower amount of data.