# OpenReview forum: "Large Pre-trained time series models for cross-domain Time series analysis tasks"
_ICLR.cc/2024/Conference — Submitted to ICLR 2024_

### Official Review · Reviewer_e2Vd · 2023-10-26

**Soundness:** 2 fair
**Presentation:** 2 fair
**Contribution:** 1 poor
**Rating:** 3
**Confidence:** 5

**Summary:**

This paper introduces a time series pre-training framework that leverages the concepts of patching and masked token reconstruction, both of which have been extensively studied and utilized in time series modeling. The authors specifically put forth an adaptive patch resampling aimed at better aligning time series patterns across various domains. While this paper is well-structured in general, the core contributions and technical novelty appear to be constrained. Some assertions within the manuscript lack sufficient evidence (refer to my detailed comments below). Additionally, the overall presentation could benefit from further refinement. On the experimental front, related & important baselines are absent and the main experiemntal setting is ill-defined. While time series pre-training holds potential and merits exploration, I believe this work requires substantial improvements before it is fit for publication.

**Strengths:**

- The motivation for time series pre-training is well established; I agree with the authors regarding the overall narrative.
- The proposed patch resampling is technically feasible, and the ablation studies demonstrate the effectiveness of this design.
- The overall pre-training & fine-tuning pipeline is well structured, with two important time series analytical tasks (i.e., forecasting and classification) undertaken.

**Weaknesses:**

- The overall technical novelty is limited. The primary contribution of this work lies in the concept of patch scoring, as presented in Eq. 2, and the subsequent two paragraphs. The overarching design can be seen as an extension of PatchTST. In the realm of time series pre-training, several recent studies, such as SimMTM, have delved into the concept of masked patch reconstruction.

- The experimental settings are ill-defined. While this work emphasizes cross-domain adaptation, the evaluation datasets (& domains) substantially overlap with the source datasets (& domains) used in pre-training. I do not think this is a valid evaluation protocol for cross-domain adaptation.

- The presentation could benefit from further refinement. For example, Fig.1 offers limited information, and upon examining just this figure and its caption, I have several related questions unsolved. Furthermore, numerous claims and technical assertions are not adequately backed by evidence or in-depth discussion. Please refer to the questions I've enumerated below for further clarity.

**Questions:**

**Questions & Detailed comments**

1. I question the validity of the claim, "... unlike text data, each individual timestamp may not provide enough semantic meaning about local temporal patterns of the time series." In natural language processing, doesn't a single token also sometimes fail to convey the full semantic information of a sentence?

2. Regarding the construction of the pre-training set, how is the optimal combination of data samples from different domains determined? I found no discussion on this in the experiment section.

3. How can Eq.2 effectively handle "out-of-distribution" samples from domains that were not encountered during pre-training?

4. I'm not sure about what the authors intend by "how good" in the paragraph following Eq.2.

5. In section 3.3, what are the fundamental differences between random masking and last token masking in time series pre-training? Is there any deeper analysis or extended discussion available?

6. Fig1 is confusing. After reviewing Fig1 and its caption, I have several related inquiries: Q1: Why choose GRU over a linear projection as the patch embedder? Q2: How do patch scoring and pruning operate? Q3: What do you intend by h(1) to h(R)? Q4: Where is the self-supervised optimization highlighted?

7. Several critical baselines, such as PatchTST, SimMTM, and TimesNet, are absent. Additionally, it would be advantageous to evaluate using datasets from domains not encountered during pre-training.

---

> ### Author Response · Authors · 2023-11-22
> **Response to Review**
>
> We thank the reviewer for their comments. We address their questions as follows:
>
> **In natural language processing, doesn't a single token also sometimes fail to convey the full semantic information of a sentence?**
>
> We specifically refer to semantic information of each individual token. While single tokenized word/subword does not impart full semantic meaning, they have inherent semantic meaning of varying importance. In contrast, for time-series, each individual time-step is a real number that doesn’t provide good semantic interpretation of local temporal dynamics in isolation in most cases.
>
> ** how is the optimal combination of data samples from different domains determined?**
>
> Our main aim is to show that LPTM can adapt to pre-train and learn from wide range of domains. Therefore, we do not optimize on the right combination of datasets to optimize downstream performance. We chose widely used datasets from time-series literature that covers diverse range of applications.
>
> However, finding or optimizing for the right combination of datasets is an interesting research direction which actively explored even in LLM pre-training and can be explored for time-series as well using LPTMs.
>
> **How can Eq.2 effectively handle "out-of-distribution" samples from domains that were not encountered during pre-training?**
>
> The segmentation module identifies good segmentation strategies for each of the domains encountered in the pre-training. Therefore, they cannot be directly used on downstream tasks of new domains. However, they can handle unseen datasets and tasks of domains trained pre-training.
>
> **I'm not sure about what the authors intend by "how good" in the paragraph following Eq.2.**
>
> The scoring function is trained to predict the “goodness” of segments by helping find the subset of segments that decreases the SSL losses during pre-training. This is later explained in Section 3.4 (Eq. 4) where scoring function s(i,j) is trained to be similar to negative log of SSL losses which determined how good the chosen set of segments are in performing the SSL tasks.
>
> **In section 3.3, what are the fundamental differences between random masking and last token masking in time series pre-training? Is there any deeper analysis or extended discussion available?**
>
> Random masking masks a small fraction of tokens chosen at random for each batch. Last masking masks the last few tokens only and aims to predict it. Random masking helps the model to learn to generate missing parts of the time-series at any time-period and helps learn contextual temporal dynamics is relation to observed time-series. Last masking is more similar to forecast that determines on the parts of the time-series in the future. Both the losses help model understand and generate time-series from different perspectives and help improve efficacy of LPTM for most benchmarks.
>
> **Regarding Fig 1**
>
> To model the score of segments we need to capture the temporal dynamics of the segment which is done by GRU. This is then also used to embed the segments for the same reason. Moreover, a linear projection cannot captured segments of varying length and their dynamics. The patch scoring is done using the scoring function (Eq 2) and the high scoring subset of segments is chosen and pruned by heuristic mentioned in Sec 3.2 after Eq. 2. We also define h(i) in the paragraph as the the maximum scoring segment that start from time-step i. The full model in Figure 1 is pre-trained on the SSL tasks (Sec 3.3) to learn the parameters.

---

> > ### Comment · Reviewer_e2Vd · 2023-11-23
> > **Response to the rebuttal**
> >
> > Thank you for addressing some of my questions. However, most of my concerns remain unsolved. At this point of time, I intend to keep the current score unless the authors directly address the listed weaknesses and the 7th question.

---

> > > ### Author Response · Authors · 2023-11-23
> > > **Response to weaknesses and additional baselines**
> > >
> > > We would gladly reply to the weaknesses of the reviewer as follows as detailed in other responses and additional results:
> > >
> > > **Regarding technical novelty**
> > >
> > > The segmentation module is indeed the main novelty of the paper. However, we would like to emphasize that learning domain-specific adaptive segmentation strategies is vital for multi-domain pre-training. Therefore, we cannot simply use PatchTST or other variants of time-series input for all domains. Indeed we have compared LPTM with PatchTST in the new experiments (here)[https://openreview.net/forum?id=KJ1w6MzVZw&noteId=LqihMFfTsp] where LPTM clearly outperforms PatchTST is all benchmarks. While we used simple pre-training methods that have been previously studied, more sophisticated ones can indeed be designed on top of LPTM to improve its performance further.
> > >
> > > **the evaluation datasets (& domains) substantially overlap with the source datasets (& domains) used in pre-training. I do not think this is a valid evaluation protocol for cross-domain adaptation.**
> > >
> > > LPTM segmentation module learns segmentation strategies for each domain and can enable evaluation on unseen datasets and tasks on unseen datasets but on domains observed during pre-traning. Therefore, LPTM doesn;t support adaptation to novel unseen domains. However, it is the first of its kind to learn to adapt to unseen datasets across all the pre-train domains leads to the development of a general pre-trained model across multiple domains.
> > >
> > > We also provided additional clarification on Fig 1 in previous response.
> > >
> > > **Question 7: Missing baselines**
> > >
> > > We have added additional baselines of more state-of-art forecasting time-series models including TimesNet and PatchTST at (here)[https://openreview.net/forum?id=KJ1w6MzVZw&noteId=LqihMFfTsp] and show that the performance of LPTM is superior in electricity, traffic, and M3 benchmarks and similar to SOTA in other benchmarks.

---

### Official Review · Reviewer_mzJa · 2023-10-30

**Soundness:** 2 fair
**Presentation:** 1 poor
**Contribution:** 3 good
**Rating:** 5
**Confidence:** 5

**Summary:**

The paper proposes a cross-domain/dataset self-supervised learning approach to pre-train a time series model. They perform masked reconstruction with a Transformer architecture, introducing a dataset specific segmentation module to transform time series data into intermediate representations which are subsequently fed into the Transformer model. They pre-train the model on 7 datasets from various domains, and evaluate on forecasting and classification tasks.

**Strengths:**

This paper tackles the ambitious problem of cross-domain/dataset pre-training for time series to learn a general model for time series tasks. They successfully pre-train such a model across a variety of datasets, and show decent performance across tasks and datasets.

**Weaknesses:**

1. Writing can be greatly improved. Abstract should be 1 paragraph. Mathematical notation is not clear -- many variables are not defined.
2. Empirical comparisons are somewhat lacking. More recent baselines can be included (PatchTST, TimesNet for forecasting, CoST for self-supervised forecasting). More evaluation metrics can be presented (MAE, sMAPE, ...).
3. The usefulness of the model is diminished with the dataset specific segmentation module. The model is unable to perform zero-shot forecasting or prediction tasks.
4. Codebase in given link is incomplete. No script for training / predictions. README is empty, without instructions.

**Questions:**

-

---

> ### Author Response · Authors · 2023-11-22
> **Response to Review**
>
> **Regarding Writing**
>
> We thank the reviewer for their comments. We would be grateful if the reviewer can point to specific instances where the writing is unclear.
>
> Out abstract has multiple paragraphs to first introduce the novel challenges of pre-trained time-series models that are not yet well explored and then introduce the technical novelties and advantages of LPTM in subsequent paragraphs. The abstract is less than 200 words and clearly introduces the scope, importance and novelty of our work.
>
> To the best of our knowledge we have clearly defined all the mathematical notation in the paper. The only source of notational confusion, as other reviewers pointed out is that we have one type in Eq 4 where $g(i,j)$ should be $s(i,j)$.
>
> **Regarding zero-shot forecasting**
>
> Pre-trained LPTM models are observed to perform similarly or better than SOTA baselines using significantly smaller training data and training time. However, we need some amount of fine-tuning to specific task since each task is different to pre-training SSL tasks and require the model to adapt to it.
>
> However, designing zero-shot learning on LPTM pre-trained models is indeed a very interesting future research problem with non-trivial challenges we do not tackle in our current work.
>
> **Codebase**
>
> We thank the reviewer for the comments. We will improve the documentation of the code when we release it publicly.

---

### Official Review · Reviewer_KRyA · 2023-10-30

**Soundness:** 4 excellent
**Presentation:** 2 fair
**Contribution:** 3 good
**Rating:** 5
**Confidence:** 3

**Summary:**

This paper introduces a novel method for pre-training time series models to be used in a wide range of downstream tasks.
The method relies on a segmentation of the input time series into possibly overlapping segments that are further encoded using self-attention.
A segment selection strategy is used to focus on most informative parts of the time series, and automatically select segment lengths.

**Strengths:**

The proposed method relies on a simple yet interesting idea that is to extract important segments of varying lengths from time series such that each segment will be treated the way a token is processed in standard NLP pipelines.
The high-level presentation of the method is rather accessible (though the technical details are much harder to grasp due to problems with the notations, see below) and the experiments tend to validate the choices that are made.

**Weaknesses:**

There are many mistakes in the notations that make it hard (or even impossible) to fully grasp what is done at places.
Below is a list of such issues:
* I do not understand the rationale behind Eq (4)
    * Why taking the log of the SSL loss? If the SSL loss tends to 0 (which is probably what one targets at the end of training), then its log will have large gradient values, hence leading to unstable training
    * What is $g(i, j)$ in Eq. 4? Do you mean $s(i, j)$? If so, why summing $\log(\mathcal{L}_{SSL})$ with the sum of scores?
    * Why does the score rely on $z_i$ (is it $z_i$ or $z^{(i)}$ by the way?) and $z_j$ but not the full sequence of $z$ between indices $i$ and $j$, since recurrent units are known to have hard time catching long-term dependencies (even GRU units, to some extent)
    * In Sec. 3.2, you write:
        > The score s(i, j) for a subsequence from time-stamp i to j denotes how good the given segment is for the dataset.
        * I do not understand this sentence. What does ``how good'' mean in this context?
        * Also, given that the loss that is optimized operates on the aggregation of all scores, it is not clear how it could enforce large scores for selected segments
* The use of $h(i)$ in Fig 1 is misleading, since it looks like $h(i)$ is the hidden representation for the $i$-th segment whereas in the text $h(i)$ is said to be the index of the last timestamp for the segment starting at time $i$, and it is said that some of these segments are pruned out, hence indices of the remaining segments should not be adjacent.

All these notations should be fixed for the reader to be able to understand the technical details of the paper.

**Questions:**

Some questions are asked in the "Weaknesses" section, below is a list of additional ones:
* The text refers to "aggregation" but not much is said (in Sec 3.2 at least) on which aggregation function is used, why?

* How does your method compare to state-of-the-art methods (ie. ROCKET, COTE variants, etc.) that do not use pre-training on the given classification tasks?

>  While retrieving the optimal S(y(1...t)) is an interesting combinatorial optimization problem, [...]

Could you elaborate a bit more on this interesting combinatorial problem, does it have known solutions? Do you have a way to assess if your approximation is a reasonable one or not?

---

> ### Author Response · Authors · 2023-11-22
> **Response to Review**
>
> **Rationale behind Equation 4**
>
> Empirically we found that adding the log to SSL losses provided for much stabler training, without which pre-training was not possible. While ideally, the SSL losses converge towards zero, in practice we found that they were generally much larger throughout the pre-training process. Using adaptive loss functions across different ranges of SSL loss values could further lead to improved stability.
>
> Yes, Equation 4 does indeed have a type where $g(i,j)$ should be replaced by $s(i,j)$. We thank the reviewer for pointing this out and hope this should clarify the confusion.
>
> **Why does the score rely on z_i and z_j but not the full sequence of
>  between indices and ...**
>
> Yes, GRU may fail to capture important long-term dependencies. However, the alternative of combining all the indices in the sequences may be challenging since we need to still present the ordering of the sequence of hidden representations to make sure the model can view the time-steps i to j in order. Further addition of multiple segments may make the segmentation module inefficient in finding scores for all possible segments.
> However, improving on this drawback of capturing all segments' patterns effectively and efficiently is an interesting problem for future work.
>
> **Implication of score function**
>
> The score function $s(i,j)$ is trained to predict the logarithm of SSL losses (Eq 4) which enables the model to in turn use it to find better segmentation strategies. This is because a higher $s(i,j)$ using the trained segmentation module implies the segment is useful for the model to undestand important patterns and properties of the time-series (via the SSL tasks).
>
> **h(i) in the Fig 1**
>
> The scoring function first determines the indices $h(i)$ for each time-step $i$ and we use it to prune the set of segments to input to the transformer model. We agree that that not all the segments caan be chosen and some will be pruned. We will update the figure to reflect this.
>
> **which aggregation function is used, why?**
>
> The aggregation of the time-step representations $z_i, \dots, z_j$ refers to simple summation after passing through a self-attention layer. We will clarify that. We used summation since it is a standard aggregation function used to aggregate along the temporal dimension to represent a time-series, or a segment (in our case) using a fixed length vector embedding.
>
> **Comparison with ROCKET and other baselines**
>
> The TARNet baseline we compared against provided significantly better performance that these methods as shown their paper. Therefore, we only compred against TARNet which is the state-of-art classification model.
>
> **About the combinatorial problem**
>
> The combinatorial problem referred to is to find the optimal set of segments that cover the entire time-series that have the maximum sum of scores. The set of satifiable soultions is exponential in the number of time-steps. We used a simpler heuristic that guarantees the coverage of the segments but doesn't guarantee optimality. A more thorough theoretical analysis of the problem would be an interesting problem to improve the performance of LPTM.

---

> > ### Comment · Reviewer_KRyA · 2023-11-22
> > **Answer to your feedback**
> >
> > Thank you for your feedback.
> > Rather than writing you will make the changes in the paper, it would probably have been more convinving to actually update the paper so that the reviewers can make their minds about the updated version of the paper.
> >
> > > The score function is trained to predict the logarithm of SSL losses (Eq 4)
> >
> > Do you mean that minimizing the loss in Eq 4 is supposed to lead to the sum of scores of selected segments tending to the log of the MSE? I do not see why this is something we would want (once again, for low SSL losses, the log is negative, hence you would penalize large segment scores, which is counter-intuitive).

---

> > > ### Author Response · Authors · 2023-11-22
> > > **Clarification for Eq 4**
> > >
> > > Yes. We apologize for the confusion. There is a typo in Eq. 4. The equation is $\mathcal{L}_{g} =\Sigma_S (g(i, j) + \log \mathcal{L}_SSL)^2$. The only change is the addition on square of right hand side. This loss ensures the the score function is close to negative log of SSL loss. Thus larger score implies lower SSL loss.

---

### Official Review · Reviewer_jLeR · 2023-10-31

**Soundness:** 2 fair
**Presentation:** 1 poor
**Contribution:** 2 fair
**Rating:** 3
**Confidence:** 4

**Summary:**

The paper proposes a model which can be pre-trained on multiple time-series from diverse domains and can perform a wide range of tasks. Their proposed model is trained by masking a proportion of time segments. The authors argue that uniform length segments cannot scale across datasets and tasks.

**Strengths:**

The paper attempts to pre-train a model which can solve multiple tasks on multiple time-series from diverse domains. To the best of my knowledge, this is amongst the first few studies which attempts to do this, demonstrating promising performance.

I also appreciate that the authors compare their methods with some domain specific baselines.

**Weaknesses:**

I would encourage the authors to address the following to improve their paper:

1. **Reproducibility:** While the code is available, many hyper-parameters important to reproduce the results are not mentioned in the manuscript. (1) As an example, the authors do not mention the proportion of time segments that they mask during self-supervised learning ($\gamma$). (2) The number, length, ranges, number of channels etc. of time-series use for pre-training and evaluation are not mentioned either. (3) Furthermore, the authors compare "training time (minutes) till convergence" but fail to mention the compute infrastructure, what kind of time (wall clock?) are they measuring.
2. **Clarity:** The paper is unclear and many statements are not rigorously or scientifically define. For e.g., (1) the authors claim in Section 3.2, that their segment score function measures "how good the given segment is for the dataset", but do not clarify what the notion of goodness is? The notion of goodness is also not immediately clear as the authors use a hyperbolic Tangent function as the scoring function. (2) The authors invoke $g(i, j)$ for the first time in Equation 4. Consequently, it appears to me it seems that the paper was put together in a hurry, without careful proof-reading. Also see Questions.
3. **Claims:** The authors claim that variable sized segmentation is a key contribution of their work, but they only compare with time-step level segmentation. While they cite PatchTST, they do not compare with fixed length time-series segmentation, and hence it is unclear whether the contribution leads to significant gains over what seems to work (i.e. uniform time-series segmentation).
4. **Baselines:** Some state-of-the-art forecasting baselines are missing, e.g., PatchTST and TimesNet from ICLR 2023, along with statistical forecasting methods such as AutoARIMA, AutoTHETA, AutoETS, Naive etc., and non-transformer-abed deep learning methods such as N-HITS and N-BEATS.
5. **Experimentation:** (1) A pre-trained model should be able to solve tasks without any fine-tuning, especially since all the training parts of the datasets are observed during pre-training. (2) For smaller datasets, a large model trained from scratch is destined to under fit. Since the authors have not mentioned the size of the model, beating LPTM trained from scratch on a small dataset can be attributed to the model being too big for a small dataset. A smaller model might very well learn from scratch.

Minor:
1. Please fix the capitalization of the datasets. The diseases should be capitalized.
2. Pease fix the citations using \citep or \cite, and \citet, whichever is appropriate.

**Questions:**

1. What is the size of the model? How many layers of transformer? What is the number of heads? What is the size of embeddings?
2. What is $g(i, j)$ in Equation 4?
3. What is $\gamma$? How are the segments sampled?
4. What are the key differences between this work and "PEMs: Pre-trained Epidemic Time-Series Models."?
5. See questions in Weaknesses

**Details Of Ethics Concerns:**

There seems to be a lot of overlap with another paper that I had reviewed for NeurIPS 2023, which is also concurrently submitted to ICLR 2024 -- PEMs: Pre-trained Epidemic Time-Series Models.

At least one paragraph is completely copied -- please see the paragraphs in Section 3 of this paper and PEMs on Linear-probing and fine-tuning.

I suspect there is a lot more overlap.

---

> ### Author Response · Authors · 2023-11-22
> **Response to Review**
>
> We thank the reviewer for their comments.
>
> **Reproducibility**
>
> We provide additional details on important hyperparameters in Appendix Section B.
>
> >  For RANDMASK, we found the optimal gamma = 0.4, and for LASTMASK
> gamma = 0.2 was optimal
>
> For Flu-US we pre-trained and trained on the dataset from 2004-2012 and evaluated for the 2013-2020 seasons. We used the standard recommended train-val-test split for ETT2, PEM-Bays, NY-Bike, Nasdaq, and M3 as done in other works. The ratio is usually around 70-20-10. The test and training data splits are also provided for classification datasets from the source UCI repository.
>
> We also mention the computing environment in Appendix Section B as using Nvidia Tesla v100 GPU with 32 GB VRAM. The CPU used is an Intel Xeon with 64 cores and 128 GB of memory. The code running time is benchmarked using wall time or execution of the entire pipeline of training.
>
> **Clarity**
>
> The score function $s(i,j)$ as defined in Equation 2 provides a score for each of the time-series segments that is used to pick the appropriate segments of the model. This is optimized in Equation 4 by making the score function reflect the negative of SSL training loss. Therefore, set of segments with higher scores will provide lower SSL loss.
>
> Equation 4 does indeed have a type where $g(i,j)$ should be replaced by $s(i,j)$. We thank the reviewer for pointing this out and hope this should clarify the confusion.
>
>
> **On requirement of fine-tuning**
>
> Even pre-trained language models or vision models required some fine-tuning or prompting to adapt to a downstream task. Therefore, it is not clear why a pre-trained model does not need fine-tuning. Indeed, the SSL tasks differ significantly from the downstream tasks such as classification and forecasting. While LASTMASK has similarities with forecasting, the horizon length required for forecasting differs for each downstream task the the last segment length of the segmentation module.
>
> **Smaller models may outperform LPTM**
>
> We do not believe that smaller models trained from scratch can outperform LPTM. Indeed some of the baselines like EpiFNP, MICN, and ColaGNN which are not transformer-based have a smaller number of parameters but in most cases underperform LPTM significantly and only slightly outperform LPTM is specific benchmarks in few instances.
>
> **Differences with PEM work**
>
> This work and the submission "PEM: Pre-trained Epidemic Time-series Models" are very different and do not have any overlap:
>
> 1. PEM paper focuses on generating pre-trained epidemic time-series models that are exclusively used for epidemic analysis tasks such as forecasting, epidemic onset and peak detection, etc. This work tackles a much more general problem of pre-training models for generic time-series from different domains.
> 2. The novel contributions in PEM paper are mostly on novel SSL methods that are specifically designed for epidemic time-series. This work does not focus on any novel SSL task but instead introduces a novel segmentation module that can enable models to pre-train from datasets of multiple domains including and beyond epidemic time series.
>
> While both works tackle challenges for pre-training time-series models, their approaches and scope are vastly different.

---

> > ### Comment · Reviewer_jLeR · 2023-11-22
> > **Response to Rebuttal**
> >
> > Dear Authors,
> > Thanks for your response.
> >
> > I agree with Reviewer KRyA that making changes to the manuscript directly would have definitely helped.
> >
> > Having said that, I do not agree with the following:
> > 1. "pre-trained language models or vision models required some fine-tuning or prompting to adapt to a downstream task." -- Large language models can be used in a zero-shot setting, and so can vision-language models like CLIP. There is even work on zero-shot forecasting using meta-learning, so I do not think that zero-shot experiments are far fetched.
> > 2. "We do not believe that smaller models trained from scratch can outperform LPTM." -- I think it would have been nice to demonstrate this belief experimentally. Nonetheless, I still think that for small datasets, smaller models might actually do better. I am happy to cite research on forecasting on the same.
> > 3. "This work and the submission "PEM: Pre-trained Epidemic Time-series Models" are very different and do not have any overlap" -- There is significant overlap in the way the papers are phrased, notations, and the contributions are marginally different in my opinion.
> >
> >  Given these considerations, I would like to stick to my score.

---

### Official Review · Reviewer_xd4P · 2023-11-01

**Soundness:** 2 fair
**Presentation:** 2 fair
**Contribution:** 2 fair
**Rating:** 3
**Confidence:** 5

**Summary:**

The paper introduces an effective framework for pre-trained time series models and demonstrates strong empirical performance on diverse forecasting and classification tasks. The adaptive segmentation technique is a key contribution enabling learning from heterogeneous time series data.

**Strengths:**

Originality:

The idea of pre-training time series models on diverse datasets from multiple domains is highly original and innovative. This enables knowledge transfer and improves efficiency similar to language and vision domains.

The adaptive segmentation module for handling diverse time series dynamics during pre-training is a creative technique and novel contribution.

Clarity:

The paper is clearly structured and easy to follow. The problem context, proposed method, experiments and results are presented logically.

Technical details are clearly explained and intuition behind design choices is well-articulated.

Tables and graphs effectively summarize key quantitative results.

Significance:

This work makes important strides towards general pre-trained models for time series, which might have high impact if the quality is good enough.

The ideas could inspire more research into techniques for pre-training on diverse time series data.

**Weaknesses:**

This paper has some obvious limitations which may lead the reviewer tend to reject it:

The model architecture used is quite straightforward - just a transformer encoder. Exploring more sophisticated temporal modeling architectures could be beneficial.

More in-depth analysis into the effect of pre-training like how the adaptive segments evolve could provide useful insights.

Ablations only evaluate the removal of components, could also analyze additions like other SSL tasks.

Hyperparameter sensitivity analysis is limited - how do factors like segment score thresholds affect performance?

Though diverse, the pre-training datasets are still limited to a few domains. Expanding the data diversity could help.

Theoretical analysis on how pre-training and adaptive segmentation provide benefits is lacking.

Comparisons to more sophisticated domain-specific models like those using additional covariates would be informative.

Analysis of computational requirements for pre-training is needed, especially regarding scaling up.

Testing on a wider range of time series analysis tasks like anomaly detection could help show broad utility.

Lack of analysis of any negative societal impacts or limitations of the approach.

Lack of baselines: for PEMS-Bays and METR-LA, we have STGNN, StemGNN, GraphWavenet and so on; for ETT dataset, we have PatchTST, FEDformer. Timesnet and so on. The lack of such important baselines makes this paper hard to position.

The word of "multi domain" is overused: the reviewer don't see the specific module for multi domain setting. However, the "a separate segmentation module for each dataset domains to capture varied sizes of segments that differ across datasets" in page 3 Section 3.1 limits the ability of generalization on this model.

**Questions:**

See Weekness.

**Details Of Ethics Concerns:**

-

---

> ### Author Response · Authors · 2023-11-22
> **Response to Review**
>
> We thank the reviewer for the comments. We would address some of them as follows:
>
> **The model architecture used is quite straightforward - just a transformer encoder. Exploring more sophisticated temporal modeling architectures could be beneficial.**
>
> Our important contribution is the importance of learning good segmentation strategies for enabling effective representation for multi-domain time-series pre-training.
> The main challenge for performing pre-training on heterogeneous datasets
> is not the model architecture but the segmentation module.
> We show that by using the vanilla transformer architecture we generally outperform state-of-art models designed for specific downstream tasks.
> Our segmentation module can be applied to any transformer-based architecture to improve and enable multi-domain pre-training
>
>
>
> **Ablations only evaluate the removal of components, could also analyze additions like other SSL tasks.**
>
> Many SSL tasks are usually designed for specific applications or for pre-training on a single data source. Moreover, they may not automatically
> be retrofitted with our framework of using segments of time series.
> Moreover, we showed significant performance improvement using simple SSL tasks of random masking and last token prediction.
> Identifying better SSL tasks that can be performed on segment tokens is an interesting future research direction.
>
> **Though diverse, the pre-training datasets are still limited to a few domains. Expanding the data diversity could help.**
>
> We leverage datasets from seven very diverse domains ranging from traffic and finance to epidemiology and behavior tracking. Moreover, these datasets are popular benchmarks widely used in literature.
> Therefore, we showed the efficacy of LPTM in pre-training on diverse datasets. If the reviewer can suggest any other specific domains we would gladly consider adding them as well.
>
> **Theoretical analysis on how pre-training and adaptive segmentation provide benefits is lacking**
>
> While we empirically show the efficacy of the segmentation module in handling diverse datasets, generating a convincing theoretical justification is non-trivial.
> This is due to the complexity of the model and dataset distributions from different domains. Our main motivation is based on various patterns of smoothness and local dynamics of the time series that cannot be captured by tokenizing single time stamps or uniform length segments.
> This is also analogous to various tokenization strategies used in large language models where the strategies are not theoretically justified but are motivated by empirical and linguistic arguments.
>
>
> **Comparisons to more sophisticated domain-specific models like those using additional covariates would be informative.**
>
> This is an interesting question. LPTM architecture is designed to be a general time-series forecasting model without additional sources of features similar to other top-performing state-of-art baselines. Adapting our methods to multiple covariates is an important challenge for future work.
>
> **Analysis of computational requirements for pre-training is needed, especially regarding scaling up.**
>
> We provided the computing environment and time for pre-training and training in Appendix Section B. We also measure the compute time required for training in Table 3 and show that LPTM requires a significantly smaller time to reach the performance of best baselines.
>
>
>
> **Lack of analysis of any negative societal impacts or limitations of the approach.**
>
> We comment on the potential negative societal impacts as discussed on page 9 Section 6 such as the requirement of testing for fairness of predictions, data quality issues, and ethical impacts of the decisions taken by the forecasts.
> While a detailed analysis of ethical impacts is not the scope of our work, we agree that understanding such impacts is an important problem.
>
>
>
>
> **The word of "multi domain" is overused: the reviewer don't see the specific module for multi domain setting. However, the "a separate segmentation module for each dataset domains to capture varied sizes of segments that differ across datasets" in page 3 Section 3.1 limits the ability of generalization on this model.**
>
>
> The segmentation module's main goal and main contribution of our work is to identify segmentation strategies for datasets from each of the different domains.
> This allows LPTM to effectively pre-train and deploy on datasets of these domains.
> Therefore, we aptly emphasize the ability of LPTM to handle multi-domain data sets and tasks.

---

> > ### Comment · Reviewer_xd4P · 2023-11-23
> > **Thanks for the response**
> >
> > Dear authors,
> >
> > Thanks for the reply! Your answers have addressed my major questions. However, according to the general score and the common issues identified by all the reviewers,  I tend to keep my score.
> >
> > Best

---

### Author Response · Authors · 2023-11-22
**Additional baseline comparison**

Based on suggestions on most reviewers we performed additional experiments on other baselines as follows:

|           | Flu-US | Flu-japan | ETT1 | ETT2 | PEM-Bays | NY-Bike | NY-Taxi | Nasdaq | M3    |
|-----------|--------|-----------|------|------|----------|---------|---------|--------|-------|
| LPTM      |   0.79 |       704 | 0.39 | 0.46 |      2.5 |    2.37 |   11.84 |   0.17 | 0.872 |
| TimesNet  |   1.12 |      1056 |  0.4 | 0.45 |      3.9 |    3.14 |   13.55 |   0.24 |  1.55 |
| PatchTST  |   1.32 |      1135 | 0.41 | 0.44 |      3.7 |    3.22 |   12.56 |   0.29 | 0.943 |
| STGNN     |   1.14 |      1459 | 0.51 | 0.64 |      2.5 |    2.64 |   10.33 |   0.24 | 1.231 |
| N-HITS    |   1.47 |      1054 | 0.47 | 0.52 |      3.4 |    2.94 |   12.66 |   0.24 | 1.155 |
| AutoARIMA |   2.17 |      1344 | 0.73 | 0.64 |      4.1 |    4.13 |   16.33 |   0.62 |  1.89 |

The overall superiority of LPTM in electricity, traffic, and M3 benchmarks remains. Further, the performance of LPTM is almost similar to SOTA baselines in other benchmarks as well.

---

### Meta-Review · Area_Chair_2fdX · 2023-12-09

**Metareview:**

This paper tackles an important task of pretraining a general time series model from diverse datasets (from different domains). A novel contribution of this paper is the adaptive segmentation module which can adaptively learn the segmentation strategy (to generate the time series segments as tokens). Some of the concerns on this paper (e.g., writing, clarity, baselines, in-depth analysis, etc.) were shared across the reviewers. Overall, I agree on these issues. The authors spent great efforts on rebuttal, and 4 out of 5 reviewers actively participated in the discussions. However, the discussions did not help much to change the ratings. Considering that no reviewers support the acceptance, my recommendation is to reject this paper.

**Justification For Why Not Higher Score:**

Some of the concerns on this paper (e.g., writing, clarity, baselines, in-depth analysis, etc.) are shared across different reviewers. Meanwhile, no reviewers support the acceptance - the scores from all the 5 reviewers are negative.

**Justification For Why Not Lower Score:**

N/A.

---

### Decision · Program_Chairs · 2024-01-16

Reject